# Multilayer Network Approach in EEG Motor Imagery with an Adaptive Threshold

**DOI:** 10.3390/s21248305

**Published:** 2021-12-12

**Authors:** César Covantes-Osuna, Jhonatan B. López, Omar Paredes, Hugo Vélez-Pérez, Rebeca Romo-Vázquez

**Affiliations:** Departamento de Bioingeniería Traslacional, CUCEI, Universidad de Guadalajara, Guadalajara 44430, Mexico; cesar.covantes@alumnos.udg.mx (C.C.-O.); jhonatan.lopez@alumnos.udg.mx (J.B.L.); omar.paredes@academicos.udg.mx (O.P.); hugo.velez@academicos.udg.mx (H.V.-P.)

**Keywords:** adaptive threshold, coherence, functional connectivity, multilayer network, otsu

## Abstract

The brain has been understood as an interconnected neural network generally modeled as a graph to outline the functional topology and dynamics of brain processes. Classic graph modeling is based on single-layer models that constrain the traits conveyed to trace brain topologies. Multilayer modeling, in contrast, makes it possible to build whole-brain models by integrating features of various kinds. The aim of this work was to analyze EEG dynamics studies while gathering motor imagery data through single-layer and multilayer network modeling. The motor imagery database used consists of 18 EEG recordings of four motor imagery tasks: left hand, right hand, feet, and tongue. Brain connectivity was estimated by calculating the coherence adjacency matrices from each electrophysiological band (δ, θ, α and β) from brain areas and then embedding them by considering each band as a single-layer graph and a layer of the multilayer brain models. Constructing a reliable multilayer network topology requires a threshold that distinguishes effective connections from spurious ones. For this reason, two thresholds were implemented, the classic fixed (average) one and Otsu’s version. The latter is a new proposal for an adaptive threshold that offers reliable insight into brain topology and dynamics. Findings from the brain network models suggest that frontal and parietal brain regions are involved in motor imagery tasks.

## 1. Introduction

The brain is a complex system with spatio-temporal dynamics that can be mapped by techniques that measure brain activity: electroencephalography (EEG), magnetoencephalography (MEG), and functional magnetic resonance imaging (fMRI) [1]. These techniques have been widely used to model brain networks that represent the structural and functional connectivity of the brain. Among all those techniques, EEG is an accessible, widespread method that measures the electrical activity of the brain on the scalp with a time resolution in milliseconds [2]. EEG analyses have divided brain waves into five major frequency bands: delta, δ (0.5–4 Hz); theta, θ (4–8 Hz); alpha, α (8–13 Hz); beta, β (13–30 Hz); and gamma, γ (30–128 Hz) [3]. Network models based on these frequency bands have revealed distinctive patterns and brain dynamics that have been used to study both normal and pathological mental states [4,5,6]. These network models can be analyzed using graphs built from an adjacency matrix that results from a brain connectivity analysis.

Brain connectivity analyses estimate the interaction strength among local information processing areas of the brain. Current state-of-the-art reports three types of connectivity: structural, based on the anatomical structure of the brain; functional, that measures the statistical dependence of different brain areas; and effective, which estimates causal relations among brain regions [7]. Concerning functional connectivity, literature describes various methods of estimation; including correlation (time domain dependence), and coherence (frequency domain dependence) [7]. Coherence measures the statistical relationship between two signals in the frequency domain [8] and it has been widely used in cerebral activity analyses involving memory [9], mathematical [5,6], and reasoning [10] task studies. It has also been applied to analyze differences at specific frequencies in patients with brain disorders [11], such as Parkinson’s [12] and Alzheimer’s diseases [13] and epilepsy [14]. In this work, adjacency matrices calculated from coherence between brain areas in electrophysiological bands were used to estimate functional connectivity.

Motor imagery is a cognitive-motor process widely studied by coherence analysis that has the potential to trigger and control actuators in brain-machine interface systems without any external motor action. Such systems aim to control a device through the brain activity of a user. Recent studies have focused on characterizing EEG through graph analysis to pinpoint not only brain areas but also interactions between them [15].

A graph is a mathematical tool used to describe the brain as a set of nodes (brain regions) and edges (connections) [16]. In Graph theory, there are different kinds of graphs, among which we can mention single-layer and multilayer ones. In single-layer networks, the edges represent the same type of connections between nodes. The associations between zones depend on a single character, which may be directed or undirected [17]. Some studies of brain connectivity have examined the brain as a single-layer graph linked by a single temporal or frequency property [18,19]. In cases where nodes can be linked based on multiple characters, associations are treated independently to build multiple single-layer networks that ignore the synergy between characters. Multilayer networks are suitable for these scenarios because they have the flexibility required to integrate multiple types of interactions in a single model.

The brain is currently considered a multilayer network [20]. As it was pointed out by [21], brain networks are intrinsically multilayers. There is not a single neuronal connectivity pattern able to fully represent brain functioning. Then, a multilayer framework is suitable for analyzing brain connectivity without either throwing away or combining different information. This focus improves understanding of brain complexity and interaction spectra with no need to discard electrophysiological data. This approach has proved to be a powerful tool in describing the complex organization and evolution of the human brain and its relationship to cognition [22]. Multilayer networks have been applied in brain analyses [23] using fMRI [24,25], MEG [4,26], gene expression [27] and EEG [20,21,28,29] techniques. The range of topological properties to be explored is, therefore, wider than in classic single-layer modeling [30]. Here, the efficiency of information flow results from multilayer interdependence within the network, rather than being an effect of each layer individually [31].

In the workflow of graph analysis, a common practice consists of thresholding networks to eliminate spurious connections [32]. That is because functional connectivity analysis, through measuring the statistical dependence among brain areas, yields a continuous weight range for interaction strength. Since some of these interactions should be labeled as spurious by the randomness of the signal, it is critical to exclude them from the brain connectivity analysis.

In this study, two thresholds were tested: the fixed (average) threshold, which is widely used in the literature, and a recently proposed threshold called Otsu. The fixed threshold method establishes a single, absolute threshold value over the entire network, typically fixed by averaging the adjacency matrices [33]. Values above this average are considered connections and are assigned a value of 1, while values below the average are discarded and receive a value of 0 that results in a binarized adjacency matrix. The main disadvantage of this approach is that a fixed threshold based on averages is conditioned by the weight distribution in the adjacency matrices, but this means that it will behave unreliably in the presence of outliers and non-normal distributions.

In contrast to the fixed threshold, Otsu’s approach involves optimizing the threshold value by evaluating how well the binarization process identifies two types of data (i.e., pixels, voxels, etc.) [34]. Some applications of Otsu’s methodology include structural segmentation in fMRI [35,36,37,38], and noise removal in EEG recordings using wavelet decomposition [39]. In our case, Otsu’s methodology was implemented for image segmentation and binarization [40]. To the best of our knowledge, and after an exhaustive literature search, Otsu’s method has not been applied to estimate the threshold of adjacency matrices in brain connectivity analyses. In this context, and considering the adjacency matrices as images that contain information about brain connectivity gathered from EEG recordings, this work proposes to apply Otsu’s threshold to these matrices to estimate an optimal threshold for brain connectivity analyses.

In light of the foregoing, this study aimed to analyze EEG dynamics by classical single-layer and multilayer network models for a motor imagery dataset. This was conducted to feature the movement and its dynamics, and thus pinpoint patterns capable of feeding a BCI system. The coherence adjacency matrices for each electrophysiological band (δ, θ, α and β) of the brain areas were analyzed individually on a single-layer approach, and then integrated, considering each band as a layer, to build a brain network model following the multilayer approach. Both approaches were built with fixed and Otsu’s thresholds.

Our results show that multigraph models cluster the four studied movements and lead to pinpointing the key electrodes for the motor imagery task that are located mainly on the frontal and parietal cortex. These brain zones coincide with the results presented in [15,41,42,43,44]. These works model brain connectivity with single-layer approach and a known threshold. However, our work explores a proof-of-concept EEG multilayer brain connectivity with an adaptative threshold. For this purpose, the paper is organized as follows: Section 2 addresses the material and methods, including the database description, the EEG signal preprocessing, and the connectivity estimation; in Section 3 the threshold, and single-layer and multilayer networks approaches are introduced, concluding with the results and discussion of the single-layer and the multilayer brain models with both thresholds in Section 4. The paper ends with the conclusions.

## 2. Materials and Methods

### 2.1. Database

In this study, the open access BNCI Horizon 2020 dataset (2a of BCI Competition IV) [45] was retrieved to pinpoint patterns of motor imagery. This dataset consists of 18 EEG recordings (Figure 1a) taken from 9 subjects (recorded in two sessions on different days) for four different motor imagery tasks (Figure 1b): left hand (class 1), right hand (class 2), feet (class 3), and tongue (class 4). The signals were recorded at a 250 Hz sampling rate and then band-pass filtered between 0.5–100 Hz. Electrodes were placed according to the 10–10 International System at Fz, Fc3, Fc1, Fcz, Fc2, Fc4, C5, C3, C1, Cz, C2, C4, C6, Cp3, Cp1, Cpz Cp2, Cp4, P1, Pz, P2, and POz.

The experimental paradigm for each trial is illustrated in Figure 1c [46]. On the trials, subjects began by focusing their eyes on a black screen (*t* = 0 s). After two seconds *t* = 2 s, an arrow image pointing left, right, down, or up (representing one of the four classes) appeared and remained on the screen for 1.25 s. Subjects then carried out the corresponding motor imagery task until the arrow image on the screen disappeared at *t* = 6 s, indicating a brief pause before the beginning the next trial. The time window corresponding to motor imagery (MI) onset) *t* = 3.5–5.5 s of the experimental paradigm was analyzed.

### 2.2. Preprocessing

Each EEG recording was composed of 6 runs (Figure 2a) separated by a short break. Each run consisted of 48 trials (12 for each class), resulting in 288 total trials of 2 s each (72 for each class).

To reduce the EEG spatial interference, a Common Average Reference (CAR) filter (Equation (Equation 1)) was applied for each of the 288 two-second EEG windows.
(1)$$V_{i}^{CAR}=V_{i}^{CR}-\frac{1}{N}\sum_{j=1}V_{j}^{CR}$$
where $V_{i}^{CR}$ represents the potential between electrode *i* and the reference electrode, and *N* is the total number of electrodes.

Once the 288 windows filtered, each two-second window was transformed into the frequency domain. The power spectral for the 72 windows of each motion class was averaged to obtain 4 two-second frequency-averaged EEG windows. This process was carried out on each of the 18 recordings (Figure 2b).

### 2.3. Connectivity Estimation

The coherence index values between two signals range from 0 to 1. A value close to 1 indicates a strong relationship, while a value close to 0 represents weak interactions between signals. Coherence index is defined as (Equation (Equation 2)):(2)$$C_{xy}\left(f\right)=\frac{{\left|S_{xy}\left(f\right)\right|}^{2}}{S_{xx}\left(f\right)S_{yy}\left(f\right)},$$
where *x* and *y* are two signals or channels, $C_{xy}\left(f\right)$ is the coherence spectrum matrix as a function of a given frequency *f*, $S_{xy}\left(f\right)$ is the cross-power spectrum, and $S_{xx}\left(f\right)$ and $S_{yy}\left(f\right)$ are the auto-power spectra of *x* and *y*, respectively [47].

## 3. EEG Processing

### 3.1. Layers Construction

To generate the single- and multilayer network models for the used motor imagery dataset, four layers were estimated, each corresponding to the main electrophysiological bands (δ, θ, α and β). Each layer was built by estimating the coherence among the EEG electrodes, then averaging the magnitude of the frequencies that comprised each band. This approach generated an adjacency matrix for each band (Figure 2c).

As mentioned above, four two-second averaged windows were obtained from the 18 EEG recordings for each MI class. After that, functional brain connectivity was estimated in each window by calculating the pairwise coherence indices among the 22 electrodes. This allowed us to obtain 22 × 22 weighted adjacency matrices for each class as a layer. Figure 3 shows an example of a β band-coherence adjacency matrix for the left-hand IM. Red indicates a high coherence value, while blue represents weakly connected areas. These layers were evaluated by the approaches of a single layer, where the layers of electrophysiological bands were analyzed separately; and the multilayer, where each class layer was integrated to build a multiple network model.

### 3.2. Threshold Estimation

The threshold stage (Figure 2d) is a key step in graph analysis that provides reliable estimates of the network topology [48] and preserves the local topological features of the network measures [16,49]. In this study, the adjacency matrices were thresholded to build the connectivity networks using two methods: the widely used fixed threshold approach (i.e., average degree across the network) [33], and a proposal for a novel method based on image segmentation the Otsu’s method [40].

#### Otsu’s Threshold

This threshold uses the adjacency matrix data to calculate data distribution represented as a histogram Figure 4). In brain networks, histograms such as this one correspond to the scores of the weighted adjacency matrix. In our case, the matrices consisted of 22 × 22 values from 0 to 1 (coherence range values).

For example, if we fix the threshold at *T* = 0.01, then adjacency values below *T* can be classified as class $C_{1}$ and correspond to spurious connections. Values above *T* are classified as class $C_{2}$ and correspond to effective connections. Thus, connections in $C_{1}$ are counted and divided by the total number of connections, *N* (22×22), to obtain the intensity $w_{1}$, and, likewise, for $C_{2}$ to estimate the intensity, $w_{2}$. The means, $\mu_{1}$ and $\mu_{2}$, and variances, $\sigma_{1}^{2}$ and $\sigma_{2}^{2}$, of these intensity values are also estimated, and the procedure is repeated for each increment of *T* until the range of values is completed. Obviously, all connections for $C_{2}$ are 1.

Next, the “Within-Class Variance (WCV)” (Equation (3)) and the “Between-Class Variance (BCV)” (Equation (4)) were computed in this threshold.
(3)$$\begin{array}{cc} WCV & =w_{1}\sigma_{1}^{2}+w_{2}\sigma_{2}^{2} \end{array}$$
(4)$$\begin{array}{cc} BCV & =w_{1}w_{2}{\left(\mu_{1}-\mu_{2}\right)}^{2} \end{array}$$

The optimal threshold is the value that minimizes WCV while maximizing BCV. Figure 5 shows an example of the distributions for an adjacency matrix with a maximum BCV and a minimum WCV. As can be seen, the optimal threshold is *T* = 0.9698. Once calculated, the weighted adjacency matrix is binarized. An example of this procedure is shown in Figure 5.

Comparing the values of the thresholds obtained by the fixed (*T* = 0.9747) and Otsu’s methods (*T* = 0.9698) we find that they tend to be similar. Therefore, the binarized matrices obtained from these thresholds (Figure 6) are close related. This suggests that both methods could generate similar topologies. However, as mentioned above, Otsu’s threshold has the advantage of estimating an optimized threshold based on the distribution of the weights in the adjacency matrix, while the fixed threshold average is sensitive to outliers and non-normal distributions.

### 3.3. Single-Layer Network Estimation

To model brain dynamics in motor imaginary tasks by the single-layer approach, multiple single-layer graphs were built for each class for all 18 EEG recordings. Each graph corresponds to a band network representation (δ, θ, α, and β) of the 22 EEG electrodes as the graph nodes, and the brain wiring or graph edges corresponding to the effective band-coherence score between electrodes. Notice that such graphs are independent one of another, despite in nature, the brain oscillome is not compartmentalized but modulates electrophysiological bands as a whole. Then, 72 graphs were obtained for each MI class that corresponds to the four frequency bands of the 18 EEG MI recordings.

Then, four graph metrics were estimated: degree (Equation (Equation 5)), that measures the electrode neighborhood by adding all *j*-column $a_{ij}$ adjacency matrix coefficients for the *i*-node *v*; eigenvector centrality $x_{v}$ (Equation (Equation 6)), that evaluates the neighborhood (M(v)) integration by estimating the eigenvalues λ and their eigenvector $x_{t}$; *k*-core number (Equation (Equation 7)), that represents the electrode coreness level where each node’s score *k* is the subgraph G(C) to which it belongs with degree nodes $d_{G(C)}\left(v\right)$ greater than *k*; and PageRank (Equation (Equation 8)), that ranks the node importance by averaging the ratio of its neighbors’ pagerank $P_{R}\left(v\right)$ and their degree d(v).
(5)$$d\left(v\right)=\sum_{i,j\in V}a_{ij}$$
(6)$$x_{v}=\frac{1}{\lambda }\sum_{t\in M(v)}x_{t}$$
(7)$$\forall v\in C:d_{G(C)}\left(v\right)\geq k$$
(8)$$P_{R}\left(v\right)=\sum_{u\in B_{v}}\frac{P_{R}\left(u\right)}{d(u)}$$

### 3.4. Multilayer Network Estimation

For the multilayer approach, the layers that correspond to each electrophysiological band were retrieve, and then integrated into multi-level graph models for each class of all 18 EEG recordings. For these graphs, the intra-layer edges were considered to be present between the nodes themselves, since all electrophysiological bands operate simultaneously. In the next step, multilayer metrics were estimated (Figure 2e,f) using the MuxViz framework in R language [50].

The metrics considered were degree, PageRank, eigenvector centrality, and *k*-core. The degree (Equation (Equation 9)) is the number of links through the layers, ignoring the interlayer link nodes themselves. PageRank (Equation (Equation 10)) is the probability of a node reaching any other node $\frac{\left(1-r\right)}{NL}$, so it ranks the nodes based on the latter probability [51]. As in a single-layer model, those probabilities are uniform, $u_{j\beta }^{i\alpha }$, through all nodes, and are interactively updated. However, in the multilayer case, the probabilities. $u_{j\beta }^{i\alpha }$, are considered to be the initial values of the next layer. For eigenvector centrality (Equation (Equation 11)), the suprajacency matrix is encoded into an aggregate matrix, $M_{j\beta }^{i\alpha }$ via an eigentensor $\Theta_{j\alpha }$. The eigenvector centrality is the dot product of the leading eigenvector, $\lambda_{1}^{-1}$ and the neighborhood of each node [52]. Finally, *k*-core (Equation (Equation 12)) represents the ratio of the coreness $n_{k-core}$ for the probability of specific degree-node $n_{k}\left(q\right)$ through all the layers [53].
(9)$$k^{i}=M_{j\beta }^{i\alpha }U_{\alpha }^{\beta }u^{j}$$
(10)$$R_{j\beta }^{i\alpha }=rT_{j\beta }^{i\alpha }+\frac{\left(1-r\right)}{NL}u_{j\beta }^{i\alpha }$$
(11)$$\Theta_{j\beta }=\lambda_{1}^{-1}M_{j\beta }^{i\alpha }\Theta_{i\alpha }$$
(12)$$P_{k}\left(q\right)=\frac{n_{k}\left(q\right)}{n_{k-core}}$$

## 4. Results and Discussion

### 4.1. Single-Layer Network

Statistical analysis for each band was performed to evaluate which electrode metric differs among the MI classes. Thus, the electrode metric distributions for each MI class were considered to be dependent variables of such class. Thereafter, a MANOVA was performed to determine the significative electrodes and followed by a post hoc test on each electrode.

MANOVA post hoc test consists of applying a one-way ANOVA on the significative electrodes and a posterior Games-Howell post hoc test, to locate the motions that have a significant difference at these electrodes.

The single-layer network results presented in Table 1 and Figure 7 show that the significant electrodes correspond to the frontal and parietal cortex in β band (Figure 7c). Post hoc analysis points that the significant electrodes on these brain areas corresponding to each graph metrics are: degree—C3, FC4, POZ, CP2 and CP3 (Figure 7d); eigenvector—C3, POZ, CP2 and CP4 (Figure 7e); *k*-core and PageRank metrics were not significative for the MANOVA. From these results, and considering those electrodes that were significative in at least two metrics, we first labeled as key electrodes: C3, POZ and CP2. Later, from these electrodes, we identified which were higher than the fixed and Otsu’s thresholds. Thus, the fixed threshold retrieved the C3, POZ and CP2 electrodes (Figure 7b), while Otsu’s threshold only retained C3 and POZ electrodes (Figure 7a).

Our results support that the frontal and parietal brain areas drive MI, as reported by Shenoy and Vinod [54]. In the latter study, the authors analyzed the same database for the four MI movements as in the present work. The common electrodes in both studies are C3, FC4, CP3 and CP4. These areas have been reported as the main MI electrodes in several connectivity analyses [15,41,42,43,44]. Most of these works are subject-wise analyzed, and their findings slightly deviate. However, all coincided with the brain zones (frontal and parietal) and the electrophysiological bands and sub-bands (mainly α and β) involved in MI.

The aforementioned picture suggests that an integrative analysis for all electrophysiological bands can retrieve the driver nodes on the MI brain dynamics. Multilayer network analysis is a model that meets the above-mentioned constraints.

### 4.2. Multilayer Network

To analyze the dynamics of MI in EEG recordings through a multilayer network model, a one-way ANOVA was performed to evaluate the multilayer metrics estimated for both thresholds. The metric distributions for each movement were obtained independently of its associated electrode; that is, all the metrics per electrode were concatenated. In this analysis, PageRank, Eigenvector and *k*-core were significatives (*p* < 0.05) for the Otsu’s threshold, while the fixed threshold in *k*-core was only significative (Table 2). This points to a difference between movements in the topology of the brain nucleus. Figure 8 shows the metric distribution for each movement.

To eliminate familywise errors, a post hoc paired *t*-test was performed using Benjamin-Hochberg FDR correction. This resulted in significative differences between the left-hand movement when *k*-core distributions were compared to the other movements for both thresholds (Figure 9).

Next, two analyses were conducted, a clustering to verify whether the movements are distinguishable based on all multilayer metrics, and statistical analysis to unveil the significant electrodes between imaginary movements.

#### 4.2.1. Clustering

To assess whether the imaginary movements diverged between them by multilayer metrics, an unsupervised approach was performed for the evaluated window data, i.e., the time from 3.5–5.5 s. Thus, all electrode metrics were concatenated and linear discriminant analysis (LDA) was carried out to lower the data dimensionality into a 3D mapping (Figure 10).

Afterward, *k*-means clustering was developed with four clusters, assuming that each cluster will depict each of the four movements. To evaluate the intersection between the estimated *k*-means cluster and the real targets (the imagery movements), the completeness score was calculated yielding a score equal to one of both thresholds.

For each threshold, the clusters mapped differently for movements. For the fixed threshold, cluster 1 (red) represents the left hand, cluster 2 (green) the right hand, cluster 3 (blue) the foot, and cluster 4 (black) the tongue. Meanwhile, for the Otsu’s threshold, cluster 1 (red) maps to the foot movement, cluster 2 (green) to the left hand, cluster 3 (blue) to the tongue, and cluster 4 (black) to the right hand. This finding points out that the multilayer graph metrics despite the threshold do illustrate the topological connectivity dynamics all during imaginary movements.

#### 4.2.2. Key Electrodes

To elucidate the electrodes most likely associated with the movements, an electrode-wise statistical analysis was conducted for all multilayer metrics among the four movements. This was evaluated by considering that the electrodes in brain dynamics are the dependent variable among the movements. In this case, a multivariate analysis of variance (MANOVA) was performed for both thresholds. The *k*-core metric was discarded in both cases since it did not comply with the MANOVA assumption that the data must be normally distributed between groups. The three remaining metrics showed significant differences (Table 3). Table 3 shows significative *p* for the eigenvector, PageRank, and degree metrics for both thresholds. This can illustrate that brain topology during imaginary movement is driven by key brain electrodes that switch to control distinct movements.

After the MANOVA analysis, a one-way ANOVA for each of the 22 electrodes was performed to identify the key electrodes that contributed to the significant differences found in the MANOVA.

These 22 one-way ANOVAs were applied for the metrics with significant *p*-value of Table 3; that is, degree, eigenvector and PageRank for both thresholds. After the one-way ANOVAs, a post hoc Games-Howell test was conducted to determine the electrodes involved in the changes in brain dynamics for the four movements. Results of this analysis are presented in Table 4.

Table 4 shows that 2 electrodes most likely drive the brain dynamics of the MI dataset analyzed. These electrodes are P2 and CP2 (Figure 11a). Among these electrodes, multilayer eigenvector metric point that the significant ones are: P2 and CP2 (Figure 11d). Degree, *k*-core and PageRank metrics were not significative for the MANOVA.

The key nodes that the fixed threshold gather the P2 (Figure 11c) electrode, while the Otsu threshold identified the electrode CP2 (Figure 11b).

These results suggests that both Fixed and Otsu’s thresholds are selective to yield the pivotal electrodes from the multilayer network. Despite that the electrodes imaged by both thresholds differ, these electrodes are neighbors and localized over the same brain area. Thus, it suggests that the Otsu threshold can recognize the underlying dynamics which widely tested thresholds as Fixed have also distinguished. Figure 12 shows an example of a multilayer graph.

The multilayer approach outlined in this study allowed us to cluster the dynamics linked to all the studied imaginary movements. Based on this finding, we pinpointed the key electrodes for such dynamics. Our results are congruent with the state-of-the-art analyses [15,41,42,43,44] that reported the frontal and parietal areas as the main brain areas in MI. In more detail, Babiloni et al. [55] indicated that sensorimotor events are correlated via the coherence with a functional coupling between parietal and central areas. All these works applied a single-layer approach for different frequency bands. To the best of our knowledge in the literature, there was not reported MI analysis based on multilayer graph models. For future work, our multilayer workflow will be tested in practical BCI applications. Our proposal, which couples an adaptive threshold with a multilayer network model, shall be cross-validated on new databases to validate its advantages over the widespread single-layer analysis.

## 5. Conclusions

In this study, we modeled single-layer and multilayer network models to analyze MI in EEG recordings. Our analysis shows that the regions activated in MI tasks are located mainly in the frontal and parietal cortex for the single-layer approach and in the parietal cortex for the multilayer approach. To pinpoint the effective connections in MI graphs a proof-of-concept threshold approach known as Otsu was proposed. The present work illustrates that combining an adaptive threshold, such as Otsu, together with integrative graph models, such as multilayer networks, produces a more reliable approximation of both the topology and dynamics associated with cognitive and motor brain functions.

Finally, future work should aim to implement this methodology to study brain connectivity in other kinds of EEG databases.

## Figures and Tables

**Figure 1 sensors-21-08305-f001:**
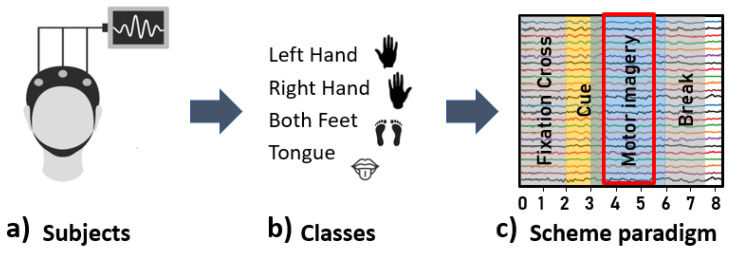
Time and scheme paradigm.

**Figure 2 sensors-21-08305-f002:**
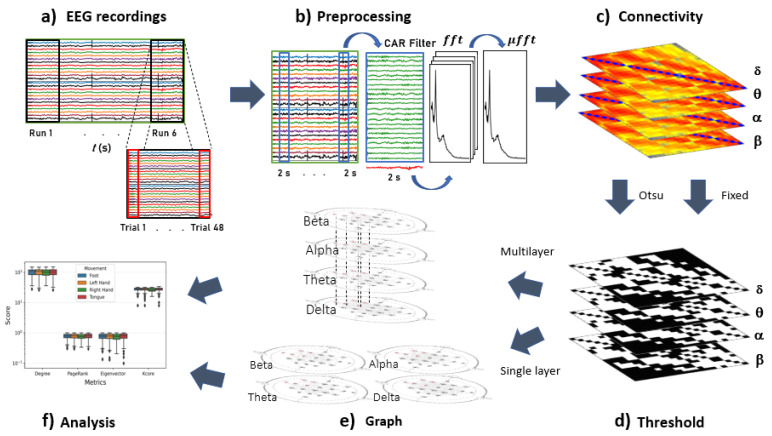
Schematic flowchart of the study methodology. Here, (**a**) correspond to the acquisition paradigm of all four classes of motor imagery, (**b**) data prepossessing, while (**c**–**f**) outline the stages of the connectivity graph analysis.

**Figure 3 sensors-21-08305-f003:**
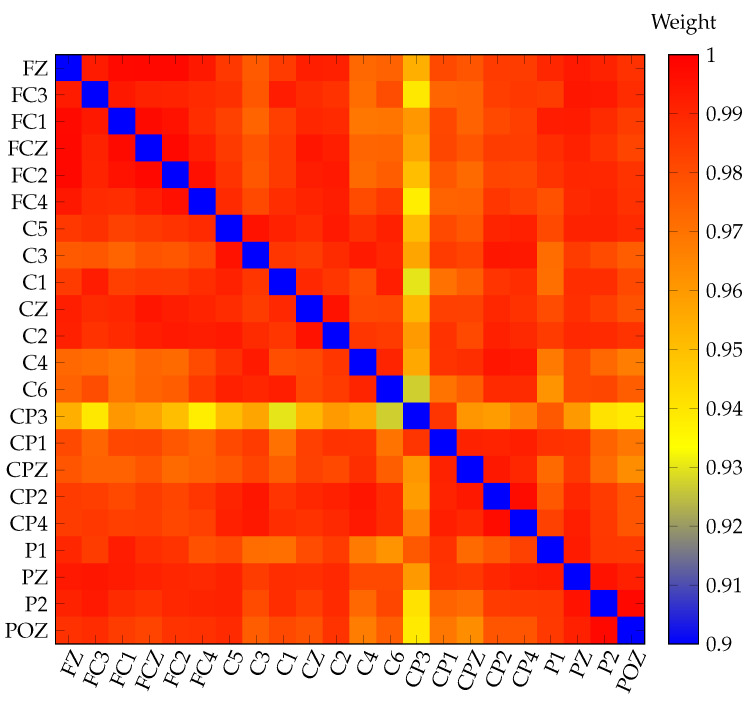
Example of a coherence adjacency matrix in the β electrophysiological band (13–30 Hz) for the IM of the left-hand.

**Figure 4 sensors-21-08305-f004:**
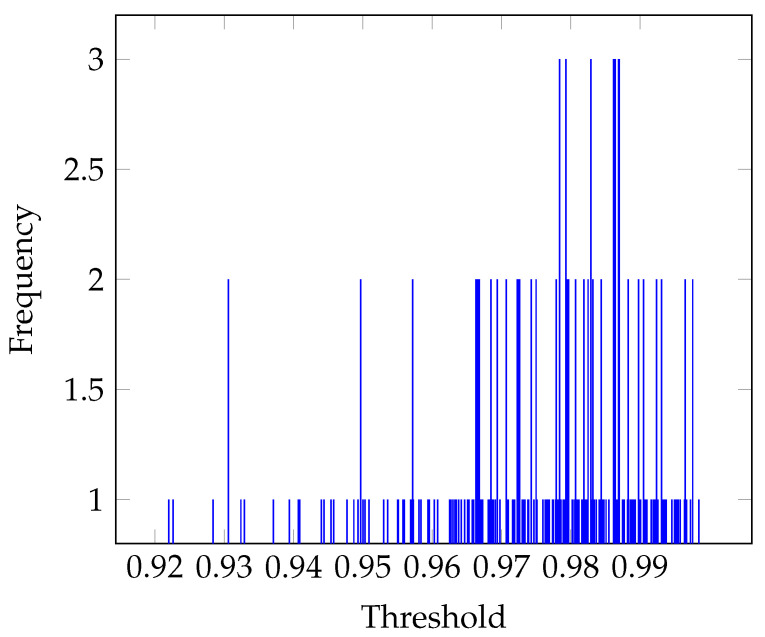
An adjacency matrix weight histogram for use in Otsu’s method. The data correspond to the β band (13–30 Hz) for the IM of the left-hand.

**Figure 5 sensors-21-08305-f005:**
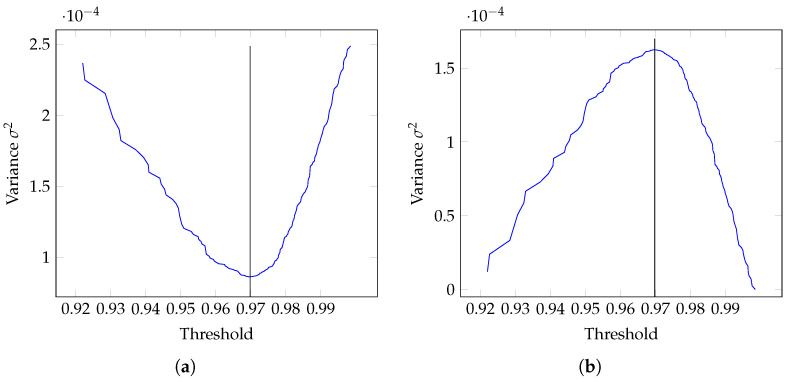
WCV and BCV histograms for the β band (13–30 Hz) for the IM right-hand. The optimum threshold value is *T* = 0.9698, indicating the minimum WCV value and the maximum BCV value. (**a**) Within-Class Variance (WCV) with the minimum value of $8.61\times 10^{-05}$ at threshold position *T* = 0.9698. (**b**) Between-Class Variance (BCV) with the maximum value of $1.62\times 10^{-04}$ at threshold position *T* = 0.9698.

**Figure 6 sensors-21-08305-f006:**
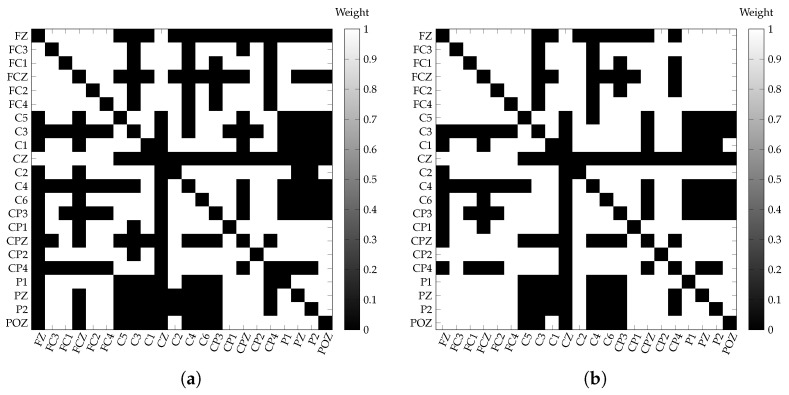
Example of the binarized adjacency matrix from Figure 3; (**a**) fixed threshold (0.9809) and (**b**) Otsu’s threshold (0.9750) for the β band (13–30 Hz) for the IM right-hand. (**a**) Example of the binarized adjacency matrix by a fixed threshold (0.9747), in the β band (13–30 Hz) for the IM of the left-hand. (**b**) Example of the binarized adjacency matrix by the Otsu’s threshold (0.9698), in the β band (13–30 Hz) for the IM of the left-hand.

**Figure 7 sensors-21-08305-f007:**
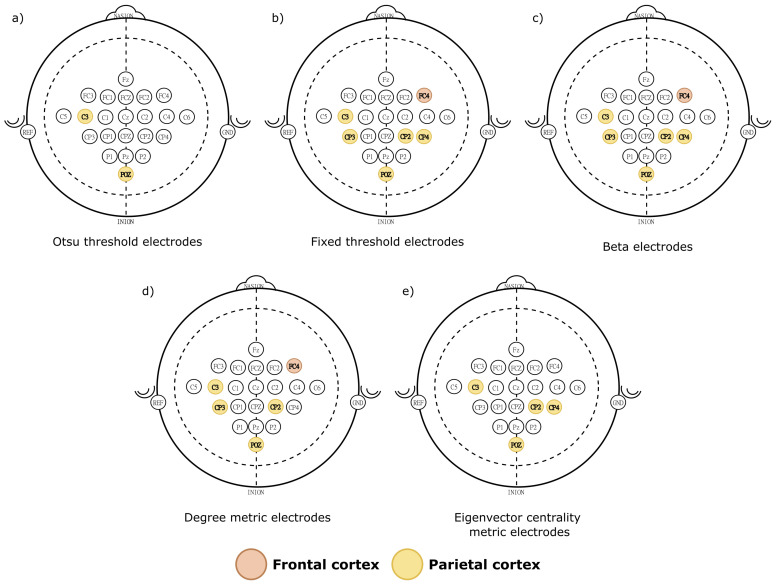
The electrodes with significative differences according to the post hoc MANOVA. In (**a**) all electrodes obtained with the post hoc analysis; (**b**,**c**) display the significative electrodes for the single-layer metrics with significative differences, while (**d**,**e**) identify the electrodes that were significative for both thresholds.

**Figure 8 sensors-21-08305-f008:**
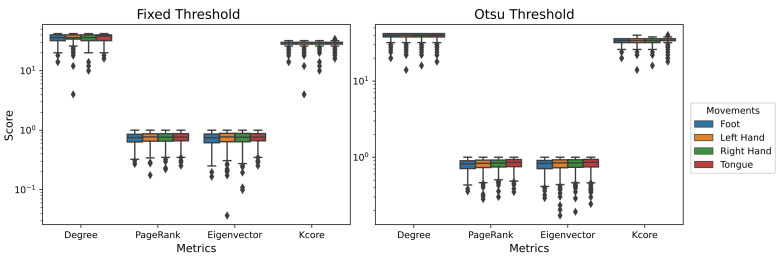
Multilayer graph metric distributions for all four MI classes. After applying a one-way ANOVA test, Otsu’s threshold showed significant values (*p* < 0.05) for *k*-core and degree, while fixed threshold only *k*-core was significantly different (*p* < 0.05).

**Figure 9 sensors-21-08305-f009:**
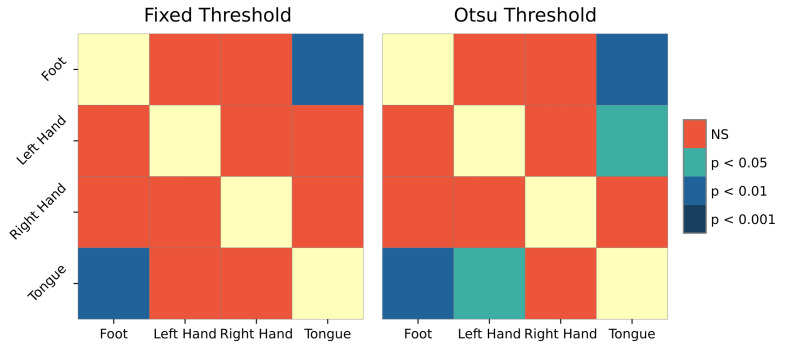
Post hoc test scores in *k*-core for Otsu’s threshold on window 3.5–5.5 s.

**Figure 10 sensors-21-08305-f010:**
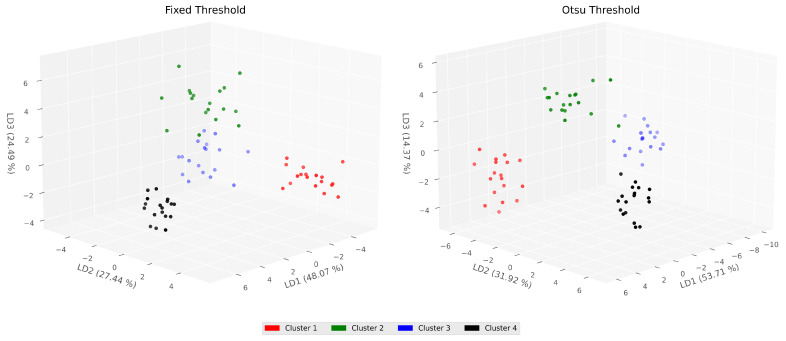
4-cluster *k*-means for a low dimensionality representation (LDA) of multilayer graph metrics, on the left for the fixed threshold and right for the Otsu’s threshold. The four estimated clusters mapped to the imaginary movements in the window 3.5–5.5 s studied in the present work.

**Figure 11 sensors-21-08305-f011:**
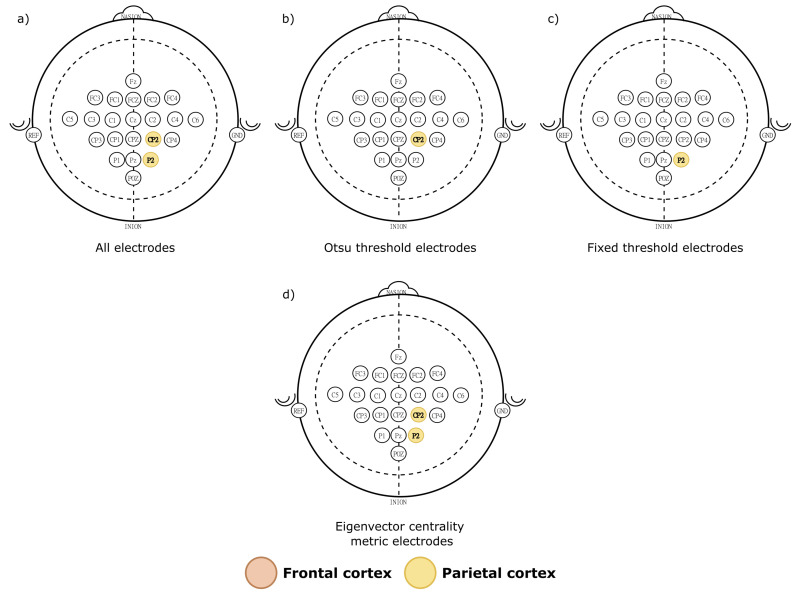
The electrodes with significative differences according to the post hoc MANOVA. In (**a**) all electrodes obtained with the post hoc analysis; (**b**,**c**) display the significative electrodes for the multilayer metrics with significative differences, while (**d**) identify the electrodes that were significatives for both thresholds.

**Figure 12 sensors-21-08305-f012:**
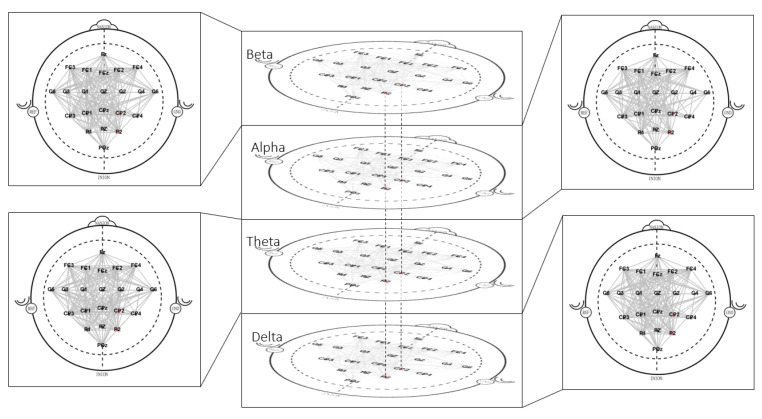
Multilayer graph for right-hand movement intention. Each electrophysiological band is a layer of the graph. Those electrodes with significative difference identified (CP2 and P2) are highlighted in red in the present study. Cross-layer edges are found for all nodes, yet only those corresponding to the significant electrodes are plotted. Each individual layer is shown separately to provide a more detailed picture of the intra-layer connectivity.

**Table 1 sensors-21-08305-t001:** *p*-values < 0.05 for the post hoc Games-Howell test in single layer of the fixed and Otsu’s threshold.

Metric	Band	Electrode	Movement	Movement	Threshold	
			1	2	Fixed	Otsu
Degree	Beta	C3	Foot	Right Hand	*	*
		POz	Right Hand	Tongue	*	*
		CP2	Left Hand	Right Hand	*	−
		CP3	Right Hand	Tongue	*	−
		FC4	Right Hand	Tongue	*	−
Eigenvector	Beta	C3	Foot	Right Hand	*	*
		POz	Right Hand	Tongue	*	*
		CP2	Left Hand	Right Hand	*	−
		CP4	Foot	Left Hand	*	−

Entries ‘−’ indicate no data according to threshold (average or Otsu); ‘*’ indicates significant data found in either the same threshold individually, or in both with the same threshold (fixed or Otsu).

**Table 2 sensors-21-08305-t002:** *p*-values from the ANOVA test on window 3.5–5.5 s.

	Fixed	Otsu
Degree	$9.42\times 10^{-01}$	$9.83\times 10^{-01}$
PageRank	$2.29\times 10^{-02}$	$3.3\times 10^{-03}$
Eigenvector	$8.72\times 10^{-02}$	$1.77\times 10^{-02}$
Kcore	$2.7\times 10^{-03}$	$3.7\times 10^{-03}$

**Table 3 sensors-21-08305-t003:** *p*-values from the MANOVA test of the average and Otsu’s thresholds on window 3.5–5.5 s.

	Fixed	Otsu
Degree	$3.72\times 10^{-01}$	$9.56\times 10^{-01}$
PageRank	$5.94\times 10^{-02}$	$6.2\times 10^{-02}$
Eigenvector	$3.48\times 10^{-02}$	$1.71\times 10^{-02}$
Kcore	Error	Error

**Table 4 sensors-21-08305-t004:** *p*-values < 0.05 for the post hoc Games-Howell test in multilayer of the fixed and Otsu’s threshold on window 3.5–5.5 s.

Electrode	Movement	Movement	Metric			Threshold	
	1	2	Degree	Eigenvector	PageRank	Fixed	Otsu
CP2	Left Hand	Tongue	−	*	−	*	−
P2	Foot	Left Hand	−	*	−	−	*

Entries ‘−’ indicate no data according to the metric (degree, eigenvector or PageRank) or threshold (average or Otsu); ‘*’ indicates significative data found in either the same graph metric individually, or in both with the same threshold (fixed or Otsu).

## Data Availability

http://www.bbci.de/competition/iv/.

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
