# Peer review of "Multilayer Network Approach in EEG Motor Imagery with an Adaptive Threshold"

_sensors, 2021, doi:10.3390/s21248305_

Round 1

Reviewer 1 Report

Review on “Multilayer network approach in EEG motor imagery with an adaptive threshold”

Description:

In this manuscript, the authors analyze EEG dynamics from one motor imagery dataset using a multilayer network modeling approach. The adjacency matrices for the four basic EEG bands (δ,θ,α,β) were used for the construction of the multi-level graph. For their construction, at first, the coherence measure was used, and then a thresholding procedure was taken place. For the thresholding the authors adopt two approaches. The first approach uses a fixed threshold, while the second approach uses an adaptive threshold using the Otsu method. The finding suggests that in motor tasks frontal and central brain’s areas are involved.

The manuscript is well written. However, there are concerns because the proposed findings are observed into one MI EEG dataset. Furthermore, these findings are well-known in BCI community. Additionally, a comparison with multiple networks, one for each EEG band must be provided to justify the usefulness of multi-layer approach.

Major Comments:

  1. Lines 47-60 : Most of reported references are irrelevant to the basic scope of the manuscript and they confuse the reader. Please remove this part of the document or at least changed it considerably.
  2. A comparison with multiple networks, one for each EEG band must be provided to justify the usefulness of multi-layer approach
  3. Lines 283 – 290 : Why finding a wider electrode diversity is a criterion of goodness? Perhaps the two electrodes describe better the underlying brain activity. It is not clear to the reviewer why the Otsu method is considered better than the fixed approach? Perhaps, some electrodes of Otsu method are spurious.
  4. Table 2 needs to be revised. Contains errors. What is ‘ ‘??

Reviewer 2 Report

This study proposes a method for brain connectivity analysis by using electroencephalography (EEG) signals. The manuscript is easy readable, but can be better organized. However, it is not clear how the results of this research can contribute to improve some scientific problems. Also, the findings should be compared better with the state-of-the-art. Other MI datasets can help to increase the impact of this study, improving the discussion. I have some comments and suggestion as follows:

Introduction

1- Cite related studies on the lines 26-28 pp.1.

2- Before presenting the objective of this research, the authors should add a paragraph to first describe the scientific problem that motivate this research, and after present the hypothesis of this current study.

3- In section Introduction, the last paragraph should further comment the novelty and relevance of this current work, with respect to the state-of-the-art.

Materials and Methods

4- In section 2. EEG processing, the authors should only present the proposed method directly, avoiding introductions when it is possible. The materials and the methodology of evaluation should be described out the section 2. Please use present tense sentences when explaining the proposed method. The current text uses indistinctly present and past tense.

5- Create a new section to describe the used datasets. In addition to the BCI COMPETITION IV DATASET 2a, the authors should add other datasets, such as BCI COMPETITION IV DATASET 2b and BCI COMPETITION III DATASET IVa, to compare better their results with the state-of-the-art. Please check that there is inconsistency with respect to the number of EEG channels used, as sometimes the text refers to 18 EEG channels or 22 EEG channels. Other inconsistency detected in the manuscript is beta band. In section Introduction is reported a frequency range from 13 to 30 Hz (see lines 25-26 pp.1), while a range from 12-30 Hz is reported in the caption of Figures 2, 3, 4, and 6.

6-In section 2.2, the first two paragraphs look an Introduction. I suggest to move both paragraphs to section 1. Introduction.

7- In section 2.3, specifically on the line 134, the authors explain that they obtained four 2 s averaged windows from 22 EEG channels. Reading the sentence I am understanding that the average calculation was performed in the time domain. Am I right? However, this processing in time domain can be sensitive to several factors, such as uncertainty related to intra-subject
reaction time variability, intra-and inter subject  skill/variability trial-by-trial when performing MI tasks, subject engagement and attention level, among others. Therefore, I recommend to first transform each 2 s window into the frequency domain to after compute the average power spectrum per channel.

8- Why a CAR filter was not applied on each 2 s window? Spatial interference, such as power line at 50/60 Hz and visual alpha (8-12 Hz) can affect very low EEG amplitudes, such as mu and beta rhythms. Please some related studies: Pfurtscheller, Gert, Christa Neuper, and Johannes Berger. "Source localization using eventrelated desynchronization (ERD) within the alpha band." Brain Topography 6.4 (1994): 269-275. McFarland, Dennis J., et al. "Spatial filter selection for EEG-based communication." Electroencephalography and clinical Neurophysiology 103.3 (1997): 386-394.

9- In section 2.4.2 the first paragraph looks an introduction. I suggest to move it to section 1. Introduction. Make a similar action with the text in the lines 77-82, which describes a novelty of this current research. Also, in section 2.5 the text on the lines 210-214 looks redundant. Please describe direclty the stage for multi-layer estimation.

10- Note that Figure 5c has the same information of Figures 5a and 5b. I recommend to edit it, avoiding redundancy.

 11- Make clear in Table 2 the meaning of some symbols, such as "*" and "-".

Results and Discussion

The authors should analyzed how their findings agree with the literature. For example, with other works that applied machine learning on the same datasets for MI classification. In this study, the brain connectivity was lateralized over the right side, in contradiction with previous studies. For instance, hand movement roduces ERD over the contralateral brain regions (around C3 right hand and C4 for left hand), while foot movement produces ERD focused on Cz. Please see below some suggestions:

12- As aforementioned, other datasets can be also analyzed for better comparison with the state-of-the-art. The findings should be compared with previous studies using the same datasets for MI recognition. In addition to other research, please see some  related studies: Ang, Kai Keng, et al. "Filter bank common spatial pattern algorithm on BCI competition IV datasets 2a and 2b." Frontiers in neuroscience 6 (2012): 39. Barachant, Alexandre, et al. "Multiclass brain–computer interface classification by Riemannian geometry." IEEE Transactions on Biomedical Engineering 59.4 (2011): 920-928. D. Milanés Hermosilla et al., "Shallow Convolutional Network Excel for Classifying Motor Imagery EEG in BCI Applications," in IEEE Access, vol. 9, pp. 98275-98286, 2021, doi: 10.1109/ACCESS.2021.3091399. Tangermann, Michael, et al. "Review of the BCI competition IV." Frontiers in neuroscience 6 (2012): 55.

13- To support better the findings, EEG channels with best connectivity can be tested in a MI recognition system.

14- Compare the results with other studies that analyzed the brain activation during upper and lower limbs MI tasks, using motor related cortical potentials (MRCPs) and/or event-related synchronization and desynchronization (ERD/ERS). In addition to other research, please see some  related studies: Pfurtscheller, Gert, and FH Lopes Da Silva. "Event-related EEG/MEG synchronization and desynchronization: basic principles." Clinical neurophysiology 110.11 (1999): 1842-1857. Hashimoto, Yasunari, and Junichi Ushiba. "EEG-based classification of imaginary left and right foot movements using beta rebound." Clinical neurophysiology 124.11 (2013): 2153-2160. Papitto, Giorgio, Angela D. Friederici, and Emiliano Zaccarella. "The topographical organization of motor processing: an ALE meta-analysis on six action domains and the relevance of Broca’s region." NeuroImage 206 (2020): 116321. Cardoso, Vivianne Flávia, et al. "Effect of a Brain–Computer Interface Based on Pedaling Motor Imagery on Cortical Excitability and Connectivity." Sensors 21.6 (2021): 2020.

16- At the end of section Results and Discussion, add a paragraph to explain the advantages, limitations, novelty, and relevance of this current study with respect to the state-of-the-art.

 Conclusion

17- The conclusion should be rewrite, focusing on the objective, application, and future work. The first paragraph looks an introduction.

References

18- This study has a total of 88 references. A total of 31(35%) references from 2017-2021, and other 24 (27%) references from 2017-2021 were used. The authors should reduce the references up 50 approximately, and update it, given priority to work  published the last five years.

Round 2

Reviewer 1 Report

The authors clearly improve the manuscript; however, additional experiments must be performed, even using only one MI dataset. These experiments could be related to the classification of MI tasks. I suggest to the authors to present experiments related to the discrimination/classification of MI trials into MI tasks (possibly by adding a new subsection in section Results).

Reviewer 2 Report

The revised manuscript has been improved substantially, and it is easy readable. The authors attended and answered the reviewer's concerns, but the proposed methodology should be revised carefully, and more technical information should be provided to facilitate the reproducibility. I have still some suggestions and comments, as follows:

  1. Something is missing in the following sentence (see lines 49-51) "Hence its potential relevance as an activation and control signal for actuators in brain-machine interface systems, which aim to control a device through the brain activity of a user. Please check it.
  2. My major concern is here. The authors affirm that the time window corresponding to the imaginary movement (IM) onset t = 2 - 4 s of the experimental paradigm was analyzed (see lines 161-163). If the authors confirm that they used this period, the results should be updated, after considering the following comment. Please note that the cue starts at 2 s and it has a duration of 1.25 s.  Thus, MI tasks were executed from from 3.25 to 6 s (t = 3.25 - 6 s). Attached I shared the document "BCI Competition 2008 - Graz data set A" for more details. It is recommended to discard the first 500 ms (from 3.25 to 3.75 s in this case) after ending the cue presentation due to event-related potentials (ERPs) that are spontaneously generated in the brain. Then, similar to other researchers, the authors should used the time interval from 3.5 to 5.5 s or from 3.75 and 5.75 s. These ERP  spontaneous can benefit the BCI performance, although the classification output could not be correlated (or related) with the user's motor intention (MI tasks). Some related references: Wolpaw J R and Boulay C B 2009 Brain signals for brain– computer interfaces Brain–Computer Interfaces (New York: Springer) pp 29–46; Pfurtscheller G and Neuper C 2009 Dynamics of sensorimotor oscillations in a motor task Brain–Computer Interfaces (New York: Springer) pp 47–64. Tangermann, Michael, et al. "Review of the BCI competition IV." Frontiers in neuroscience 6 (2012): 55.  Gouy-Pailler, Cédric, et al. "Nonstationary brain source separation for multiclass motor imagery." IEEE transactions on Biomedical Engineering 57.2 (2009): 469-478.
  3. Subsection 3.2.1 Fixed threshold is too short. The authors could consider to avoid this subsection. To facilitate the comparison, Figure 4 and Figure 7 can be merged in a unique Figure as 4a and 4b.
  4. Please explain carefully each variable used in Equations (5)-(12). The variable PR in Eq (8) looks like a product of two different variables. Please consider to use only one letter. Verify the other variable NL. Also check the representation for product operations in Equations (3) and (4). I am assuming that Equations (9)-(12) are used to present the analysis considering both alpha (α) and beta bands (β). However, it is not clear in the description.
  5. Please check that accidentally an additional parenthesis has escaped on the line 276.
  6. Could the authors to provide a pseudo-code in subsection 3.4 Multilayer network estimation to facilitate the reproducibility in future studies?
  7. Check the quality of Figures 9 and 10. Also notice that these figures are using duplicated legends (see figure 9a and 9b, and 10a and 10b), giving redundant information.
  8. Verify the product symbol in Table 2.

Round 3

Reviewer 1 Report

The  manuscript is ready for publication.

Author Response

Thank you very much! We really appreciate your comments.

Reviewer 2 Report

The revised manuscript has included in part some of my previous concerns. However, my major concern is still not solved (please see below). Then, I cannot recommend the revised manuscript for publication in high caliber journal as Sensors.

Minor comment:

1- Make clear the time interval analyzed in subsection 4.2.1.

Major concern:

2-Next description explains why the methodology and the obtained results should be revised by the authors.

On the line 50-51 the authors say "Motor imagery is a cognitive-motor process widely studied using coherence analysis and preceded by a motor intention that begins in the 0.5 – 2 s interval before the movement onset [15–17]". After, the authors confirm on the lines 142-144 that they analyzed the MI dataset over time windows corresponding to the motor intention (approximately 1 – 2 s before the imaginary movement (IM) onset) t = 2 − 4 s of the experimental paradigm was analyzed.

First of all, the studied MI dataset contains four MI classes, where the participants performed each one by following visual cues, which were display randomly. Then, I am not sure if before each cue (from 1 to 2s) there is motor planning or pre-movement activity linked to the MI task executed, as aforementioned visual cues were presented randomly.  For this reason, I think that this MI dataset is not appropriate to study motor planning or pre-movement. I recommend strongly to attend my Comment #2 of my previous report.

Please see here my comment #2:

My major concern is here. The authors affirm that the time window corresponding to the imaginary movement (IM) onset t = 2 - 4 s of the experimental paradigm was analyzed (see lines 161-163). If the authors confirm that they used this period, the results should be updated, after considering the following comment. Please note that the cue starts at 2 s and it has a duration of 1.25 s. Thus, MI tasks were executed from from 3.25 to 6 s (t = 3.25 - 6 s). Attached I shared the document "BCI Competition 2008 - Graz data set A" for more details. It is recommended to discard the first 500 ms (from 3.25 to 3.75 s in this case) after ending the cue presentation due to event-related potentials (ERPs) that are spontaneously generated in the brain. Then, similar to other researchers, the authors should used the time interval from 3.5 to 5.5 s or from 3.75 and 5.75 s. These ERP spontaneous can benefit the BCI performance, although the classification output could not be correlated (or related) with the user's motor intention (MI tasks). Some related references: Wolpaw J R and Boulay C B 2009 Brain signals for brain– computer interfaces Brain–Computer Interfaces (New York: Springer) pp 29–46; Pfurtscheller G and Neuper C 2009 Dynamics of sensorimotor oscillations in a motor task Brain–Computer Interfaces (New York: Springer) pp 47–64. Tangermann, Michael, et al. "Review of the BCI competition IV." Frontiers in neuroscience 6 (2012): 55. Gouy-Pailler, Cédric, et al. "Nonstationary brain source separation for multiclass motor imagery." IEEE transactions on Biomedical Engineering 57.2 (2009): 469-478.
